# Stochastic Processes Derive Gut Fungi Community Assembly of Plateau Pikas (*Ochotona curzoniae*) along Altitudinal Gradients across Warm and Cold Seasons

**DOI:** 10.3390/jof9101032

**Published:** 2023-10-20

**Authors:** Xianjiang Tang, Liangzhi Zhang, Shien Ren, Yaqi Zhao, Kai Liu, Yanming Zhang

**Affiliations:** 1Key Laboratory of Adaptation and Evolution of Plateau Biota, Northwest Institute of Plateau Biology, Chinese Academy of Sciences, Xining 810008, China; 2Qinghai Provincial Key Laboratory of Animal Ecological Genomics, Xining 810008, China; 3University of Chinese Academy of Sciences, Beijing 100049, China; 4Qinghai Provincial Grassland Station, Xining 810008, China

**Keywords:** community assembly, gut fungi, plateau pikas, altitudinal gradient

## Abstract

Although fungi occupy only a small proportion of the microbial community in the intestinal tract of mammals, they play important roles in host fat accumulation, nutrition metabolism, metabolic health, and immune development. Here, we investigated the dynamics and assembly of gut fungal communities in plateau pikas inhabiting six altitudinal gradients across warm and cold seasons. We found that the relative abundances of *Podospora* and *Sporormiella* significantly decreased with altitudinal gradients in the warm season, whereas the relative abundance of *Sarocladium* significantly increased. Alpha diversity significantly decreased with increasing altitudinal gradient in the warm and cold seasons. Distance-decay analysis showed that fungal community similarities were significantly and negatively correlated with elevation. The co-occurrence network complexity significantly decreased along the altitudinal gradients as the total number of nodes, number of edges, and degree of nodes significantly decreased. Both the null and neutral model analyses showed that stochastic or neutral processes dominated the gut fungal community assembly in both seasons and that ecological drift was the main ecological process explaining the variation in the gut fungal community across different plateau pikas. Homogeneous selection played a weak role in structuring gut fungal community assembly during the warm season. Collectively, these results expand our understanding of the distribution patterns of gut fungal communities and elucidate the mechanisms that maintain fungal diversity in the gut ecosystems of small mammals.

## 1. Introduction

The mammalian gastrointestinal tract is colonized by many microorganisms, most of which are bacteria and fungi [1,2]. Variations in gut microbes are closely associated with host growth and development [3], nutrient digestion and absorption [4,5,6,7], energy metabolism [8,9,10], immune regulation [11], and cognitive behavior [12,13]. Although the number of fungi in the intestinal tract is much lower than that of bacteria [1,2], they play extremely important functional roles in host metabolism and health [11,14,15,16]. For example, a previous study showed that changes in gut fungi and the interactions between gut fungi and bacteria were strongly associated with obesity in mice fed a high-fat diet (HFD) [14]. A culture-dependent method and inoculation assay with *Candida parapsilosis* in fungi-free mice suggested that the production of fungal lipases from *C. parapsilosis* is one of the mechanisms promoting HFD-induced obesity in mice [17]. A study has shown that the abundance of *Candida* in the gut of healthy people is positively proportional to the amount of carbohydrates consumed and negatively correlated with the concentration of saturated fatty acids in the gut [18]. Numerous studies have shown that gut fungi are closely associated with host disease and health [11,15,19]. For example, *Saccharomyces boulardii* is a probiotic that is widely used to block and treat bacterial infections and intestinal complications [20]. Compared to healthy individuals, the ratio of Basidiomycota to Ascomycota and the proportion of *Candida albicans* increased in the feces of patients with IBDs while the proportion of *Saccharomyces cerevisiae* decreased [21].

Many environmental factors influence the composition and diversity of gut fungi. Changes in diet composition not only have significant effects on the composition and diversity of gut bacteria but also have strong effects on gut fungi [15,22]. For example, Indian individuals have more *Candida* spp. in their feces than Japanese individuals because their diets contain a higher proportion of plant polysaccharides [22]. A high proportion of animal-related diets (e.g., cheese, meat, and eggs) significantly increased the abundance of *Penicillium* and decreased the abundance of *Debaryomyces* and *Candida* spp. in human feces, whereas a high proportion of plant-related diets (e.g., fruits, grains, and vegetables) significantly increased the abundance of *Candida* spp. [15]. The alpha diversity of the fecal fungal community of white-head cranes (*Grus monacha*) was significantly lower in the late wintering period than in the early and middle stages [23] and significantly higher in winter than in spring and autumn [24].

Many microbial diversity patterns, including latitudinal and altitudinal diversity patterns [25,26,27,28], species abundance distribution [29,30], species–area relationships [31], distance–decay relationships [32], species–time relationships [33,34], and island biological diversity distribution [35], have been observed in various ecosystems. Studies on microbial diversity in the gastrointestinal tract of chickens have shown that the fungal diversity in the upper tract is significantly higher than that in the lower tract [36]. The gut bacterial diversity of house mice changes significantly with variations in the habitat latitude [28]. The relative abundance of Firmicutes in human feces increases significantly with increasing latitude, whereas that of Bacteroidetes decreases significantly [37]. Studies on the biogeography of the gut bacterial community in plateau pika and house mice showed that microbial diversity presented an inverse altitudinal gradient diversity pattern, that is, with an increase in altitude, gut microbial diversity significantly increased [26,27]. These results suggest that the gut bacterial community displays an evident geographical distribution pattern. Fungi are an important part of the gut microbiome and coexist with bacteria in the gut ecosystem [2]. Although many studies have shown that host physiological characteristics, environmental factors, and shifts in gut bacterial taxa have significant effects on the gut fungal community [19,38,39], few studies have focused on altitudinal gradient distribution patterns and community assembly.

Community assembly explores the mechanisms that generate and maintain the stability of biodiversity. Both stochastic and deterministic processes are important for determining community assembly [40,41,42]. Vellend [43] and Stegen et al. [44] divided community assembly processes into five major ecological processes (homogenization selection, variable selection, homogenizing dispersal, dispersal limitation, and ecological drift), which simultaneously included stochastic and deterministic processes in the conceptual framework. Selection is a deterministic process that shapes community structures owing to the fitness ability of specific biological and abiotic environments that are distinct among different species, resulting in different species compositions and diversity in a given habitat community [43,44]. Variable selection, also known as heterogeneous selection, refers to the selection of species under heterogeneous environmental conditions, leading to dissimilar structures among communities [45]. Homogenous selection implies that homogeneous environmental conditions produce species selectivity, resulting in more similar structures among communities [45]. Dispersal is the active or passive migration of an individual organism from one community to another and successful colonization [43,44]. Homogeneous dispersal refers to the high frequency of dispersal or migration among communities, resulting in similar structures [44,46]. Dispersal limitation refers to the restricted migration of individual organisms among communities or successful migration but failed colonization, resulting in a greater dissimilarity among communities [44,47,48]. Ecological drift refers to fluctuations in the abundance or richness of different species in a community due to stochastic processes such as birth, death, and speciation [44,49].

Plateau pikas are typical non-hibernating small herbivorous mammals that inhabit the Qinghai–Tibet Plateau [50]. It is an endemic mammal that is widely distributed at altitudes ranging from 3100 to 5200 m.a.s.l. (meters above sea level) and latitudes ranging from 28.901° to 37.959° N on the Qinghai–Tibet Plateau [50,51,52,53]. Studies on the gut microbes of plateau pikas have mainly focused on the bacterial community and found that diet [54,55], season [5], altitude [26,56], and secondary compounds [57,58] have significant effects on the composition, diversity, and function of intestinal bacteria in plateau pikas. Additionally, a study on the effect of altitudinal gradients on the gut bacterial community suggested that bacterial diversity showed an inverse altitudinal gradient geographical distribution pattern, that is, the diversity was significantly positively correlated with altitudinal gradients, and the deterministic assembly processes of the gut bacterial community decreased with an increase in altitude [26]. Although gut fungi coexist with bacteria in gut ecosystems, the two types of microorganisms possess different biological attributes, which lead to their distinct responses to similar environmental variations [14,15,59,60]. For example, a HFD can decrease the abundance of gut fungal taxa but increase the abundance of bacterial taxa in mice [14,17,61]. In mammals, gut bacterial community diversity increases with age, whereas fungal diversity decreases [38,60]. Gut bacterial communities are more stable than fungal communities between individuals [2,15]. Many studies have focused on the gut bacterial community in plateau pikas. However, the biogeography and assembly processes of the gut fungal community in plateau pikas remain poorly understood.

To answer these questions, we collected the cecal contents of plateau pikas along an altitudinal gradient of 3118–4761 m.a.s.l. in the Qinghai–Tibet Plateau during the warm (January) and cold (August) seasons. The composition of the fungal communities in the cecal contents of plateau pikas at different altitudinal gradients was determined using ITS high-throughput sequencing. The alpha and beta diversities, relative abundance of dominant fungal taxa, co-occurrence network, and community assembly processes of the gut fungal community of plateau pikas at different altitudinal gradients were analyzed to answer the following questions: (1) Does the gut fungal diversity of plateau pikas change significantly with an increase in altitude? (2) Which dominant fungal taxa significantly increase or decrease in relative abundance with increasing host habitat altitude? (3) How important are stochastic and deterministic processes in structuring gut fungal communities along altitudinal gradients?

## 2. Materials and Methods

### 2.1. Samples

A total of 137 wild plateau pikas were captured from six different habitats in the warm (August) season of 2019 and cold (January) season of 2020 on the Qinghai–Tibet Plateau. The sample sites were the Qingshizui (3118 m.a.s.l.), Haibei (3363 m.a.s.l.), Reshui (3550 m.a.s.l.), Goulixiang (3945 m.a.s.l.), Wozhuoyi (4343 m.a.s.l.), and Kunlun Mountain passes (4761 m.a.s.l.), with altitudes ranging from 3118 to 4761 m.a.s.l. On average, ten individuals were randomly captured at each altitudinal gradient. Thereafter, pikas were euthanized and dissected, and the cecal contents were collected and frozen immediately in liquid nitrogen. Samples were then stored at −80 °C until DNA extraction. All animal experiments were approved by the Ethics Committee of the Northwest Institute of Plateau Biology (approval number: nwipb2019110801).

### 2.2. DNA Extraction and High-Throughput Sequencing

Total DNA was extracted from 137 samples using the E.Z.N.A.^®^ soil DNA Kit (Omega Bio-tek, Norcross, GA, USA) following the manufacturer’s protocol. The DNA extract was analyzed on a 1% agarose gel, and the DNA concentration and purity were determined using a NanoDrop 2000 UV-vis spectrophotometer (Thermo Scientific, Wilmington, NC, USA). The fungal *ITS* gene was amplified with the primer pairs ITS1F (CTTGGTCATTTAGAGGAAGTAA) and ITS2R (GCTGCGTTCTTCATCGATGC) using an ABI GeneAmp^®^ 9700 PCR thermocycler (Applied Biosystems, Waltham, CA, USA). The PCR amplification of the *ITS* gene was performed as follows: initial denaturation at 95 °C for 3 min followed by 27/35 cycles of denaturation at 95 °C for 30 s, annealing at 55 °C for 30 s, extension at 72 °C for 45 s, single extension at 72 °C for 10 min, with an end temperature held at 4 °C.

PCR reaction mixtures included 4 μL 5 × Fast Pfu buffer, 2 μL 2.5 mM dNTPs, 0.8 μL of each primer (5 μM), 0.4 μL of Fast Pfu DNA polymerase, 10 ng of extracted DNA, and ddH_2_O to achieve a final volume of 20 μL. PCR was performed in triplicates. The PCR products were extracted from a 2% agarose gel, purified using an AxyPrep DNA Gel Extraction Kit (Axygen Biosciences, Union City, CA, USA) according to the manufacturer’s instructions, and quantified using a Quantus™ Fluorometer (Promega, Madison, WI, USA). Purified amplicons were pooled in equimolar amounts and paired-end sequenced on an Illumina MiSeq PE300 platform (Illumina, San Diego, CA, USA) according to the standard protocols of Majorbio Bio-Pharm Technology Co., Ltd. (Shanghai, China). Raw reads were deposited in the NCBI for Biotechnology Information Sequence Read Archive (SRA) database.

### 2.3. Amplicon Sequence Processing and Analysis

Sequencing results were demultiplexed, and the resulting sequences were quality filtered using fastp (0.19.6) and merged with FLASH (v1.2.11). High-quality sequences were de-noised using the DADA2 [62] plugin in the QIIME 2 (version 2020.2) pipeline with the recommended parameters, which obtains single-nucleotide resolution based on error profiles within samples. DADA2 de-noised sequences are known as amplicon sequence variants (ASVs). To minimize the effects of sequencing depth on alpha and beta diversity measurements, the number of sequences from each sample was normalized to the minimum number of sequences represented in the total samples, which yielded an average Good’s coverage of 99.99%. Taxonomic assignment of ASVs was performed using the naive Bayes consensus taxonomy classifier in QIIME 2, and *ITS* sequencing data were analyzed using R 4.2.1.m.

### 2.4. Co-Occurrence Network

To reduce the effects of rare ASVs in the dataset, fungal ASVs present in >10% of the total samples were included in the co-occurrence network analysis. Each dot represents a fungal taxon, and the links represent robust correlations with Spearman’s correlation coefficients (r) > |0.5| and *p*-values < 0.05. The topological properties of the networks, including the total number of nodes and edges, the positive and negative edges, the ratio of negative to positive edges, and modularity, were calculated. The networks were visualized using Gephi v0.9.2 software.

### 2.5. Null Model

The mean nearest taxon distance (MNTD) computes the minimum mean phylogenetic distance between an ASV and all other ASVs in each community [63]. Standardized effect size measure (SES.MNTD) values were used to assess the assembly processes of the communities as a reflection of their phylogenetic relationships. A SES.MNTD value > +2 or <−2 indicates significant phylogenetic clustering or phylogenetic overdispersion of species in the community, and both indicate that deterministic processes dominated the community assembly. A SES.MNTD value between −2 and +2 indicates that the coexisting taxa were phylogenetically random or stochastic processes govern community assembly. The βMNTD is used to calculate the MNTD values between a given pair of samples [44]. The difference between the observed βMNTD and the mean of the null distribution is βNTI. The βNTI and Bray–Curtis-based Raup–Crick (*RC_bray_*) [44] were combined to quantify the contribution of the major ecological processes that determine the gut fungal community assembly of plateau pikas [44,64]. βNTI > +2 or <−2 indicates that variable or homogeneous selection governs community assembly, respectively [44]. If |βNTI| < +2 and RC *_bray_* > + 0.95 or <−0.95, we conclude that community assembly is governed by dispersal limitation or homogenizing dispersal [44], respectively. When |NTI| < +2 and |RC *_bray_*| < +0.95, we conclude that ecological drift contributes to community assembly [64].

### 2.6. Neutral Model

The Sloan neutral model [65] was used to determine the importance of the neutral process in assembling fungal communities. In general, the model predicts that abundant species in a metacommunity disperse between communities mainly by chance, whereas rare species are more likely to become extinct owing to ecological drift [65]. In this model, m is the estimate of the migration rate between communities. The parameter R^2^ represents the fit to the neutral model. A high R^2^ value indicates a good fit to the neutral model or a highly stochastic process assembly of the community. In this study, the datasets from each altitudinal gradient and season were used to fit the neutral model separately. The ASVs from each dataset were subsequently separated into three partitions based on the 95% confidence interval of the neutral model prediction: above-neutral (ASVs that occurred more frequently than those predicted by the neutral model), below-neutral (ASVs that occurred less frequently than predicted), and neutral (within the prediction).

### 2.7. Statistical Analysis

The software SPSS 20.0 and R (version 4.21) were used for statistical analyses. Linear regression analysis was used to explore the correlation between altitudinal gradients and the relative abundances of dominant phyla and genera, alpha diversity of the community, community dissimilarity (beta diversity within each group), and SES.MNTD and NTI values. Principal coordinate analysis (PCoA) and permutational multivariate analysis of variance (PERMANOVA) were used to evaluate the effects of altitudinal gradient on community diversity. Spearman’s correlation analysis was used to examine the correlation between the topological properties of the co-occurrence networks and altitudinal gradients. Ordinary least squares regressions were used to reveal the relationships between elevation distances and community similarities (1–dissimilarity of Bray–Curtis distance) [66]. All statistical analyses were performed using R packages, including “vegan,” “geosphere,” “igraph,” “picante”, and “Hmisc”.

## 3. Results

### 3.1. The Variation in Gut Fungal Community Composition and Alpha Diversity along Altitudinal Gradients

A total of 137 pika cecal samples were collected along altitudinal gradients in the Qinghai–Tibet Plateau during the warm and cold seasons. After amplification and sequencing of fungal microbial ITS genes, standardization, and rarified yielded 43,478 high-quality ASVs in each sample. These ASVs were grouped into 15 fungal phyla and 538 genera. The most abundant fungal phylum was Ascomycota for samples from both warm (average relative abundance, 86.30%) and cold (average relative abundance, 95.72%) seasons at all altitudinal gradients (Figure 1A,B). A significantly higher relative abundance of unclassified fungi (18.06%) was detected at 3118 m.a.s.l. (Kruskal–Wallis test, χ^2^ = 15.458, *p* = 0.009), and Glomeromycota predominated at altitudes of 3363 (15.85%) and 3945 m.a.s.l. (13.75%) (Kruskal–Wallis test, χ^2^ = 27.476, *p* < 0.001) during the warm season (Figure 1A and Appendix A). *Sarocladium* was the most abundant fungal genus in all samples (Figure 1C,D). A significantly higher relative abundance of unclassified Glomeraceae was detected at 3363 (12.28%) and 3945 m.a.s.l. (11.99%) (Kruskal–Wallis test, χ^2^ = 7.175, *p* = 0.007) during the warm season (Figure 1C and Appendix A).

Of the top five most abundant fungal phyla and genera, none of the relative abundances of the phyla displayed a significant correlation with altitudinal gradients in either warm (Appendix A) or cold (Appendix A) seasons. However, several fungal genera showed a greater correlation with altitudinal gradients (Figure 1E,F). For example, the relative abundance of *Podospora* significantly decreased with an increase in altitudinal gradients in the warm season, whereas it decreased for *Sporormiella* in both warm and cold seasons (Figure 1E,F). The relative abundance of *Sarocladium* showed a significant positive correlation with altitudinal gradient during the warm season (Figure 1E). The linear regression analysis showed that the alpha diversity, including Shannon, Chao1, and Gini–Simpson indices, significantly decreased with the increase in altitudinal gradients in both warm (Figure 2A) and cold seasons (Figure 2B), except for the Gini–Simpson index in the cold season (Figure 2B). This correlation was stronger during the warm season (Figure 2A) than during the cold season (Figure 2B).

### 3.2. Biogeographic Analysis of Fungal Communities of Plateau Pika

The PCoA ordination and similarity analysis (PERMANOVA) based on the weighted UniFrac distances showed that the composition of gut fungal communities differed significantly among altitudinal gradients during the warm (R^2^ = 0.301, *p* = 0.001, Figure 3A) and cold seasons (R^2^ = 0.171, *p* = 0.01, Figure 3B). The Bray–Curtis similarity (inter-individual similarity within each altitudinal gradient) increased with the altitudinal gradient in the warm (R^2^ = 0.243, *p* < 0.001) and cold seasons (R^2^ = 0.032, *p* < 0.001) (Figure 3C). The distance–decay relationships of Bray–Curtis dissimilarity were weak but significant during the warm season (R^2^ = 0.003, *p* = 0.016, Appendix A), cold season (R^2^ = 0.004, *p* = 0.002, Appendix A), and both seasons (R^2^ = 0.004, *p* < 0.001, Appendix A). Additionally, the results of the distance-decay relationship analysis of gut fungal communities in plateau pikas showed that Bray–Curtis heterogeneity was significantly correlated with elevation distance in the warm season (R^2^ = 0.005, *p* = 0.001; Figure 3D) and total samples (R^2^ = 0.004, *p* < 0.001; Figure 3F), but did not correlate with elevation distance in the cold season (*p* = 0.08; Figure 3E).

### 3.3. Co-Occurrence Patterns of Gut Fungal Communities along Altitudinal Gradients

To reduce the effects of rare ASVs in the dataset, ASVs present in more than 10% of the samples were included in the co-occurrence network analysis (Figure 4A–F). The topological parameters of the network, including the total node and edge numbers, positive and negative edge numbers, and the ratio of negative to positive associations between taxa, were calculated (Figure 4G–L). The total node numbers, total edge numbers, and positive edges significantly decreased with increasing altitudinal gradient (Figure 4G–I), whereas negative edges and the ratio of negative to positive associations between taxa were not significantly correlated with altitudinal gradient (Figure 4J,K). Fungal community networks increased in modularity along altitudinal gradients in the warm season, but were not significantly associated with altitudinal gradients in the cold season (Figure 4L). Overall, the complexity of the network decreased with an increase in the altitudinal gradient in both the warm and cold seasons, whereas network stability increased along the altitudinal gradient in both seasons.

Additionally, the nodes (ASVs) with higher edge numbers in the networks varied along altitudinal gradients (Figure 4A–F). Many nodes present in the networks at low altitude (3118–3550 m.a.s.l.) belong to *Sporormiella* both in warm and cold seasons (Figure 4A–C), whereas the nodes present in the networks at high altitude (3945–4761 m.a.s.l.) belong to various genera (Figure 4D–F). Although *Sarocladium* was the most abundant genus in the gut fungal community, only a few nodes were present in the network (Figure 4A–F). Interestingly, the nodes belonging to *Sarocladium* mainly interacted negatively with other taxa in all networks (Figure 4A–F).

### 3.4. Assembly Process of Gut Fungal Communities

To understand the forces that structure gut fungal community composition, we used the SES.MNTD to estimate the dataset. The results showed that the taxa in each gut community were highly phylogenetically random, as all the SES.MNTD values in each community were between −2 and +2 (Figure 5A,B), which also indicated that the gut fungal community of plateau pikas was mainly governed by stochastic processes in both warm and cold seasons. However, the importance of this stochastic process did not change along the altitudinal gradient (Figure 5A,B). To understand the forces that drive the differentiation of the gut fungal community among individuals, namely, the community assembly processes, we used the βNTI to estimate our dataset (Figure 5C,D). The results also showed that the stochastic process dominated the fungal community turnover among individuals, as most of the βNTI values were between −2 and +2 (Figure 5C,D). Additionally, the βNTI values significantly increased in warm seasons along altitudinal gradients (Figure 5C). A new ecological framework based on phylogenetic distance was used to quantify the relative contributions of major ecological processes that drive the differentiation of gut fungi among individuals. The results showed that ecological drift mainly dominated the differences in the gut fungal community among plateau pikas at all altitudinal gradients in the warm and cold seasons (Figure 5E,F). Homogeneous selection also played an important role in inter-individual differences in the group from several altitudinal gradients during the warm season (Figure 5E). Additionally, all fungal communities fit well in the Sloan neutral model (all R^2^ > 0.900, Figure 5G–L), confirming that neutral or stochastic processes played an important role in driving gut fungal community turnover among plateau pikas.

## 4. Discussion

Our results showed that several predominant gut fungal genera in plateau pikas exhibited significant correlations with altitudinal gradients (Figure 1E,F). The relative abundance of *Sarocladium* increased significantly with increasing altitudinal gradient in summer (Figure 1E). Previous studies have shown that *Sarocladium* is a predominant environmental fungal genus [67] and is also predominant in the intestinal tracts of ants and chickens [36,68] but rarely seen in the intestinal tracts of other mammals [2,17,19,69,70]. A study on the gut fungi of termites (*Nasutitermes* spp.) found that some fungal taxa belonging to *Sarocladium* were capable of degrading xylan and cellulose [68]. The relative abundance of *Sporormiella*, which is closely related to host coprophagy [71], decreased significantly with increasing altitudinal gradients in both warm and cold seasons (Figure 1E,F), indicating that the possibility of obtaining yak (*Bos grunniens*) feces gradually decreased with increasing altitude, which is consistent with the decrease in yak populations at high altitudes [50]. The relative abundance of *Podospora* decreased significantly with increasing elevation during the warm season (Figure 1E). Many species of this genus are coprophilous fungi that mainly inhabit herbivorous dung [72,73]. Therefore, plateau pikas can ingest these fungal taxa in the intestinal tract by consuming feces. Additionally, *Podospora* possesses an important role in the degradation of lignocellulose [74]. The relative abundance of *Naganishia*, an oleaginous yeast [75,76], increases significantly with altitude, producing lipids rich in monounsaturated fatty acids (MUFAs) and polyunsaturated fatty acids (PUFAs) [75,76], which can provide energy for host survival in cold environments.

The alpha diversity of the gut fungal community in plateau pikas significantly decreased with increasing altitude (Figure 2). Similar results were also found in humans, showing that the alpha diversity of oral fungi at high altitudes was significantly lower than that at low altitudes [77]. In contrast, the diversity of the gut bacteria in small mammals increases significantly with increasing altitude [26,27]. The opposite result may be due to differences in the biological features of bacteria and fungi and their preferences for different ecological niches [78,79,80]. Many studies have shown that environmental factors, such as temperature, pH, food resources, and oxygen content, can affect the abundance of dominant fungal taxa and the diversity of fungal communities [15,81,82]. Geographical variations in phenotypic and physiological features of plateau pika follow Bergman’s rule, that is, body weight and temperature increase significantly with an increase in the altitudinal gradient [83]. Compared to bacteria, fungal taxa are less suited to high-temperature environments [8,84]. An increase in altitude causes a significant increase in the body temperature of plateau pikas, which may be unsuitable for gut fungal taxa. In addition, the abundance of some gut fungal taxa displayed a significant positive correlation with dietary fiber [15,18], whereas the proportion of acid detergent fiber and lignin in plants favored by plateau pikas, such as *Elymus nutans Griseb*., decreased significantly with increasing altitude [85], which may be another potential factor contributing to the low diversity of the gut fungal community at high altitudes. Recently, several studies have shown that a large proportion of the fungal taxa in the intestinal tracts of mammals originate from the environment [36,38,86]. Many behaviors, such as ingesting food resources, digging, and consuming feces, may contribute to the migration of environmental fungal taxa attached to food sources, the soil, and feces [36,71,86]. Plateau pikas engage in coprophagy, especially during the cold season when food is scarce, and consuming yak feces is an important way for pikas to supplement their food intake [50]. However, with an increase in altitudinal gradients, the yak population gradually decreases [50], which may reduce the chance of pika consuming yak feces or the amount of yak feces consumed by pikas in high-altitude habitats. Additionally, studies on the diversity of soil and rhizosphere fungi have also shown that diversity decreases significantly with an increase in altitude [87,88].

The PCoA plots showed that the gut fungal communities of plateau pika from different altitudinal gradients were clearly separated (Figure 3A,B), and the similarity of the fungal communities within each group increased significantly with increasing altitude (Figure 3C). With an increase in elevation, there is an increase in the differentiation of environmental factors, such as temperature, food resources, and host physiological features [54,83], and both contribute to gut fungal community dissimilarity [89,90]. The concentration of atmospheric oxygen is extremely low and body temperatures of plateau pikas is very high in high-altitude habitats [27,50,83]. High body temperatures may be more suitable for the colonization and reproduction of thermophilic fungi in gut ecosystems [27], and a low-oxygen environment at high altitudes is more suitable for the survival of anaerobic fungi [82]. This may explain the decrease in the dissimilarity of gut fungi among individuals within groups in high-altitudinal habitats.

The complexity of the microbial network is closely associated with network topological parameters, including the total number of nodes and edges [91], whereas the stability of the network is more closely related to topological parameters, including modularity and the ratio of negative to positive associations between taxa [92]. Our results showed that the complexity of the gut fungal co-occurrence network in plateau pikas significantly decreased with increasing altitudinal gradient in both seasons (Figure 4A–H), and the stability of the network increased along the altitudinal gradient in the warm season (Figure 4L). Previous studies have suggested that dietary composition, exercise stress, and obesity are important factors that influence the complexity and stability of the gut microbial network [93,94,95]. The diversity of the dietary composition of plateau pikas significantly decreases with an increase in altitudinal gradients [54], which may contribute to a decrease in network complexity along the altitudinal gradients. Positive and negative associations in networks represent reciprocal and competitive relationships between connected species, respectively [92]. Increased competition among species leads to deterministic processes that construct communities [96,97]. Although the negative correlations significantly decreased along the altitudinal gradients during the warm season, the ratio of negative to positive correlations did not change in either season (Figure 4J,K), indicating that competition among the fungal taxa in the gut of the plateau pikas did not change along the altitudinal gradients.

The results of both the null and neutral model analyses support the importance of stochastic processes in shaping the gut fungal community assembly in plateau pikas (Figure 5). Similar results were found in Ujimqin sheep (*Ovis aries*), which showed that the stochastic process is more important in structuring the gut fungal community assembly than deterministic processes [80]. All the SES.MNTD values were between −2 and +2, indicating that the gut fungal community phylogenetic composition and dynamics did not differ significantly from expectations based on random community assembly [63], which suggests that the gut ecosystem of plateau pikas did not exert a strong pressure on the fungal taxa. Supporting this, many studies on gut fungi have suggested that environmental fungal taxa can randomly migrate into the gut ecosystem, mainly by ingesting food resources, and are successfully colonized [36,86]. Ecological processes, including dispersal limitation, homogenizing dispersal, and ecological drift, are stochastic forces that structure community assembly [63]. The results of the null model analysis based on βNTI and RC_bray_ values showed that inter-individual turnover in the composition of gut fungal communities in plateau pikas was mainly governed by ecological drift. Homogeneous selection (deterministic processes) had a weak effect on community assembly (Figure 5E,F). As discussed above, environmental fungal taxa can randomly migrate into the gut ecosystem, which can be identified as high ecological drift [49]. Gut microbiota, especially endotherms, occupy habitats with definable borders that have a unique ecological environment [27,98], such as high temperature and hypoxia, which is markedly different from the host survival environment. Therefore, similar environmental conditions in the gut ecosystems will select specific microorganisms. However, in our study, we found minor selective pressure (homogeneous selection) for fungal taxa in plateau pikas (Figure 5E,F). Previous studies have shown that dispersal limitation plays an important role in the assembly of the mammalian gut microbiota [32]. However, we only detected weak effects of dispersal limitation on fungal community assembly (Figure 5E,F). This is probably attributable to the fact that the plateau pikas consume not only their own and conspecific feces but also heterospecific feces, such as yak feces [50], which can eliminate the effect of dispersal limitation on gut fungal community assembly. Furthermore, regarding the gut fungal community immigration rate, m values in the cold season were higher than those in the warm season (Figure 5G–L), indicating that the dispersal ratio of fungal taxa in the cold season was higher than that in the warm season. This may be attributed to the plateau pikas engaging more in coprophagy during the cold season than during the warm season [50].

In conclusion, we found that the alpha diversity of the gut fungal community and community dissimilarity within each group significantly decreased along altitudinal gradients as well as the complexity of co-occurrence networks. Stochastic processes dominated fungal community assembly at all altitudinal gradients, and ecological drift was the main ecological process contributing to gut fungal community turnover among individuals. Highly stochastic processes govern the gut fungal community assembly, probably owing to the transportation of environmental fungi into the intestinal tract of pikas via feeding, digging, and feces consumption. Overall, these results enhance our understanding of the distribution patterns and community assembly processes of gut fungal communities in plateau pikas along altitudinal gradients.

## Figures and Tables

**Figure 1 jof-09-01032-f001:**
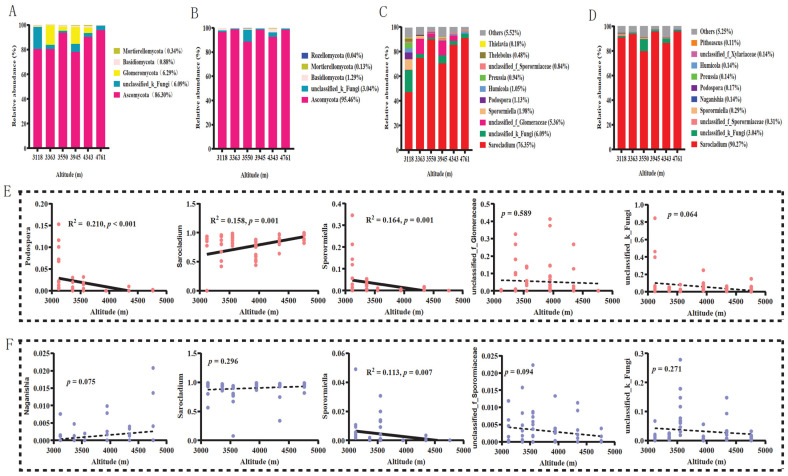
Compositions of dominant fungal phyla and genera along altitudinal gradients. The relative abundance of the top five abundant fungal phyla in warm (**A**) and cold (**B**) seasons at various altitudinal gradients. The relative abundance of the top five most abundant fungal genera in warm (**C**) and cold (**D**) seasons at various altitudinal gradients. The correlations between the top five most abundant fungal genera and latitudinal gradients in warm (**E**) and cold (**F**) seasons.

**Figure 2 jof-09-01032-f002:**
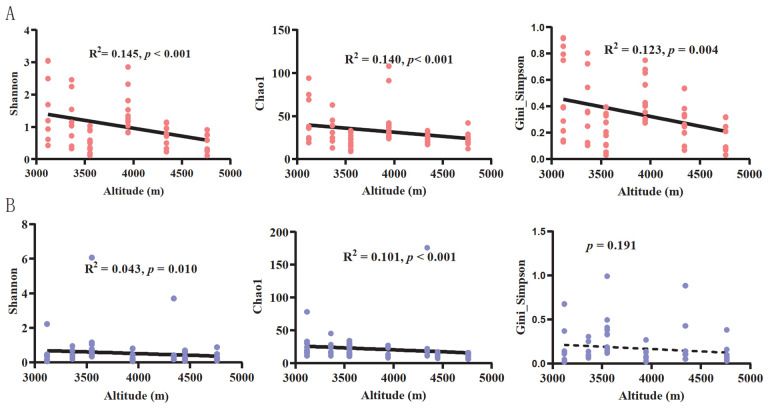
The effects of altitudinal gradients on alpha diversity of gut fungal communities. Shannon, Chao1, and Gini–Simpson indexes in relation to altitudinal gradients in the warm season (**A**). Shannon, Chao1, and Gini–Simpson indexes in relation to altitudinal gradients in the cold season (**B**).

**Figure 3 jof-09-01032-f003:**
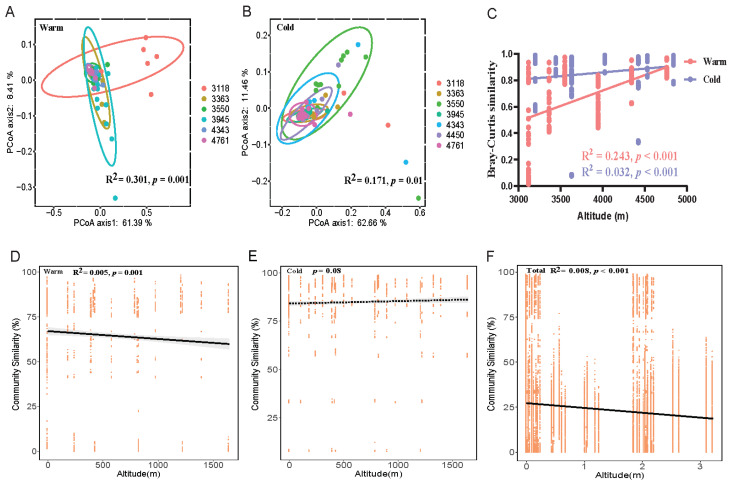
The effects of altitudinal gradients on beta diversity of gut fungal communities. Principal coordinates analysis (PCoA) illustrating community dissimilarities across elevations based on weighted UniFrac distance in warm (**A**) and cold (**B**) seasons. The beta diversity within each elevation based on Bray–Curtis distance in relation to altitudinal gradients in warm and cold seasons (**C**). Relationship between Bray–Curtis dissimilarity and elevation distance in warm (**D**), cold (**E**) and both seasons samples (**F**).

**Figure 4 jof-09-01032-f004:**
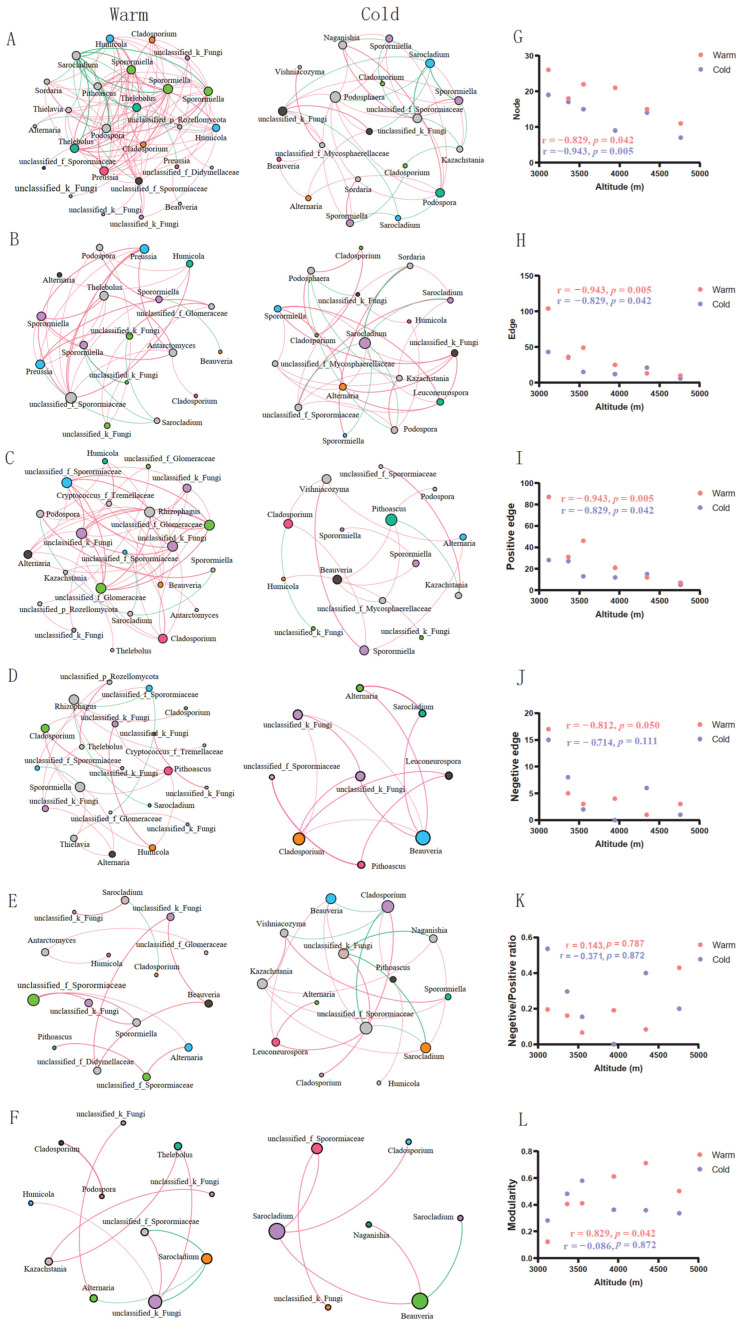
Co-occurrence network analysis of gut fungal communities across elevations. Co-occurrence networks of gut fungal community at altitudinal gradients of 3318 m.a.s.l. (**A**), 3363 m.a.s.l. (**B**), 3550 m.a.s.l. (**C**), 3945 m.a.s.l. (**D**), 4343 m.a.s.l. (**E**), and 4761 m.a.s.l. (**F**). Red and green lines represent positive and negative correlations among taxa, respectively. The size of the nodes represents the degree, while nodes belonging to one genus are indicated in the same color. The correlations between the total node number (**G**), total edge number (**H**), positive correlation number (**I**), negative correlation number (**J**), the ratio of negative to positive associations between taxa (**K**), modularity (**L**), and altitudinal gradients.

**Figure 5 jof-09-01032-f005:**
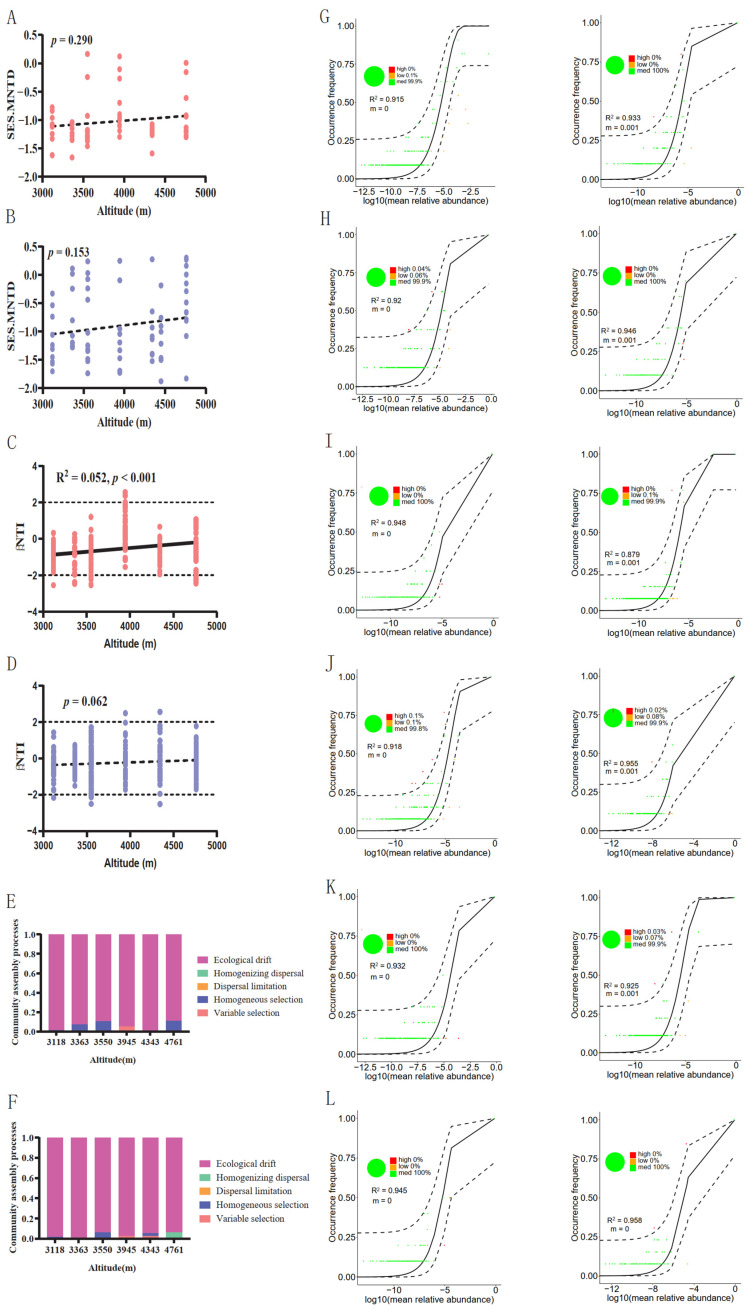
The assembly processes of gut fungal communities across altitudinal gradients. The correlation between weighted standardized effect size of the mean nearest taxon distance (SES.MNTD) and altitudinal gradients in warm (**A**) and cold (**B**) seasons. The correlation between beta nearest taxon index (βNTI) and altitudinal gradients in warm (**C**) and cold (**D**) seasons. Summary of the contribution of ecological processes in assembling the fungal communities in warm (**E**) and cold (**F**) seasons. Fit of Sloan neutral model to the gut fungal communities from altitudinal gradients of 3118 m.a.s.l. (**G**), 3363 m.a.s.l. (**H**), 3550 m.a.s.l. (**I**), 3945 m.a.s.l. (**J**), 4343 m.a.s.l. (**K**), and 4761 m.a.s.l. (**L**) in warm and cold seasons. The solid black line indicates the best fit in the neutral model, and the dashed black lines represent 95% confidence intervals around model prediction. Green circles represent ASVs that are well fitted to the neutral model. ASVs that occurred more or less frequently than predicted by the neutral model are indicated in red and orange, respectively. m indicates the migration rate and R^2^ indicates degree of fit to the neutral model.

## Data Availability

The sequence of fungal data was deposited in the Sequence Read Archive (SRA) at NCBI under the accession numbers SRP433075.

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
