# Peer review of "Stochastic Processes Derive Gut Fungi Community Assembly of Plateau Pikas (Ochotona curzoniae) along Altitudinal Gradients across Warm and Cold Seasons"

_jof, 2023, doi:10.3390/jof9101032_

Round 1

Reviewer 1 Report

Tang et al investigated the role of fungi in the intestinal tract of mammals, particularly plateau pikas inhabiting different altitudinal gradients across warm and cold seasons. This paper explores the dynamics and assembly of gut fungal communities in these small mammals. They found that the composition and diversity of gut fungi (e.g., Podospora and Sporormiella) are influenced by various environmental factors, including diet composition and altitude. Also, fungal community similarities are significantly and negatively correlated with elevation. Next, they found that the complexity of co-occurrence networks in the gut fungal community decreases with increasing altitude, suggesting changing ecological interactions. Finally, this paper reveals the importance of both stochastic (random) and deterministic (selective) processes, with ecological drift being a significant driver of community turnover. In summary, this text highlights the ecological importance of gut fungi in plateau pikas and provides valuable insights into their distribution patterns and community assembly dynamics along altitudinal gradients, ultimately contributing to our understanding of gut ecosystems in small mammals. This review paper is interesting to me. I am quite open to looking at a revised version if the authors could address some major and minor issues in a satisfactory fashion, which I describe in more detail below.

Major issues:

1.     Some figure texts seem to be horizontally compressed, e.g., Figs. 1A-D, Fig. 2, and Figs. 5E-L. Please make some changes to make the text look better.

2.     Figures 1A-D: when the authors describe the composition differences along altitudinal gradients in lines 237-247, they used the words “high”. I would suggest they perform the statistical tests whenever possible to show the significance level. Similarly, the authors may label the significantly different two compositions by adding stars (*) in panels A to D of Figure 1.

3.     Figure 4: the visualized edge weights for some panels such as panel A are too large to observe the connection between nodes. Try to reduce the edge weights so that the connectivity of networks can be better appreciated.

4.     Figures 4J&L: where are the correlation coefficients r for these two panels?

Minor comments:

1.     Lines 272-273: “based on weighted UniFrac distance” -> “based on the weighted UniFrac distance”

2.     Line 460: “select for specific microorganisms” -> “select specific microorganisms”

Only a handful of grammatical errors can be detected. Overall, it looks good to me.

Reviewer 2 Report

Stochastic processes derive gut fungi community assembly of plateau pikas (Ochotona curzoniae) along altitudinal gradients across warm and cold seasons

Dear Authors,

the manuscript is well prepared, there is not much elements to correct in the text. The main problem for me is more precise explanation of the linear regression on Figure 2 (normality of distribution and R2 value interpretation). Below I add some suggestions helpful during this process:

Line 84

In text is: Vellend [43] and Stegen [44], must be: Vellend [43] and Stegen et al. [44].

Line 104-105

In text is: altitudes ranging from 3100 to 5200 m, must be: altitudes ranging from 3100 to 5200 m.a.s.l. (meters above sea level).

Line 124, 139-141

There are meters, must be m.a.s.l.

Line 232

Space needed between (Distance-decay relationships) and [66].

Line 244-247

There are meters, must be m.a.s.l.

Line 260

Labels could be useful to read he exact value from column charts A, B, C and D or information in the legend.

Line 261

Information about the normal distribution (Shapiro-Wilk, Kolomogorow-Smirnov test) in case variables before preparing regression equation will be useful (in materials and methods).

The equation should be present in the scientific notation (Figure 1, E, first scatter plot):

There is Y=-2.367e-005* X + 0.102, should be Y=-2.367e-05*X+0.102 (in this case it is not entirely clear if power value is -5; -0.5; or -0.05). The same situation with 3rd scatterplot.

Line 266

Lack of scatter plots from  Figure 1 - F section (the altitudinal gradient in cold seasons).

Line 267

I have problem with is Figure 2, and I wondering if simple linear regression is needed there, and if R2 value doesn’t describe biodiversity (instead of linear regression).

Because when I want to predict value for altitude 3300 m.a.s.l. , then:

X = 3300

Y = -0.01*X + 4.197

Y = -0.01 * 3300 + 4.197

Y = 3.3+ 4.197

Y = 7.497

Please check if it is correct and if this R2 value doesn’t describe index for biodiversity, maybe better will be present scatterplots in this section without equation for linear regression. In case of simple linear regression R2 = 0.043 explains only 4.3% of gathered data.

Line 308-309, 316-317, 351-352

There are meters, must be m.a.s.l.

Line 510-727

References

Abbreviations of Journals should be present with dots,

ie. no. 5: Integr. Zool.

·         no. 48 – in text is Ecology letters, must be: Ecol. Lett.

·         no. 71 – in text is Quaternary Science Reviews, must be: Quat. Sci. Rev.

·         no. 78 – in text is Soil Biology & Biochemistry, must be: Soil Biol. Biochem.

·         no. 78 – in text is Acta Zoologica Academiae Scientarium Hungaricae, must be: Acta Zool. Acad. Sci. Hung.

·         no. 96 - in text is Annual Review of Ecology & Systematics, could be: Annu. Rev. Ecol. Syst.

Round 2

Reviewer 1 Report

The authors answered all my questions. I do not have further comments.